# Syntrophic Growth of *Biomaibacter acetigenes* Strain SP2 on Lactate and Glycerol

**Sofiya Parshina [1], Elena Zhuravleva [1], Anna A. Nikitina [1], Denis Grouzdev [2], Nadezhda Kostrikina [1], Vadim Kevbrin [1], Andrey Novikov [3], Dmitry Kopitsyn [3], Tatyana Kolganova [4], Roman Baslerov [4], Alla N. Nozhevnikova [1] and Yuriy Litti [1,*]**

[1] Winogradsky Institute of Microbiology, "Fundamentals of Biotechnology" Federal Research Center, Russian Academy of Sciences, 117312 Moscow, Russia; sonjaparshina@mail.ru (S.P.); zhuravleva_2595@mail.ru (E.Z.); nadin-kost@yandex.ru (N.K.); kevbrin@inmi.ru (V.K.); nozhevni@mail.ru (A.N.N.)

[2] SciBear OU, 10115 Tallin, Estonia; denisgrouzdev@gmail.com

[3] Department of Physical and Colloid Chemistry, Gubkin University, 119991 Moscow, Russia; novikov.a@gubkin.ru (A.N.); kopicin.d@inbox.ru (D.K.)

[4] Institute of Bioengineering, Research Center of Biotechnology, Russian Academy of Sciences, 117312 Moscow, Russia; moldiag@biengi.ac.ru (T.K.)

* Correspondence: litty-yuriy@mail.ru

**Abstract:** A moderately thermophilic Gram-positive chemo-organotrophic bacterium, strain SP2, was isolated by serial dilutions with crotonate and yeast extract as substrates from a butyrate-degrading methanogenic enrichment obtained from thermophilically digested sludge of the Kuryanoskaya wastewater treatment plant (Moscow, Russia). Cells of strain SP2 are spore-forming rods, sometimes occurring in short chains. The bacterium is an obligate anaerobe that grows at temperatures from 20 to 70 °C (55–60 °C optimum) within a pH range of 3.5–8 (7.5 optimum) and with NaCl concentrations of up to 2.5%. The strain utilized yeast extract and simple sugars as carbon and energy sources. Thiosulfate was used as an electron acceptor when grown on sucrose, resulting in the formation of hydrogen sulfide and the accumulation of elemental sulfur globules inside the cells. Strain SP2 is phylogenetically related to *Biomaibacter acetigenes* strain SK-G1[T] as revealed by comparison with the 16S rRNA gene (99.9% identity) and genome (ANI 99%, dDDH 90%) of both strains. It is interesting that strain SP2 was capable of syntrophic conversion of glycerol and lactate when co-cultivated with hydrogenotrophic methanogen, which was not previously shown for the SK-G1[T] type of strain. The isolation and in-depth study of new facultatively syntrophic microorganisms is important for wastewater treatment ecotechnologies due to their ability to switch to an alternative source of carbon and energy and therefore greater resistance to changing environmental conditions in bioreactors.

**Keywords:** anaerobic digestion; thermophilically digested sewage sludge; syntrophic growth; lactate; glycerol; *Thermosediminibacterales*; *Tepidanaerobacteraceae*; *Biomaibacter acetigenes*

## 1. Introduction

The problem of the disposal of organic wastes is relevant due to the increasing volumes that are produced. One of the most environmentally friendly methods of processing organic wastes is anaerobic digestion (AD), a technology that results in the production of methane-rich biogas and biofertilizers. AD is a microbiological process consisting of four main stages: hydrolysis, acidogenesis, acetogenesis and methanogenesis, with the products of the previous stage being the substrates for the next stage [1].

In well-functioning anaerobic bioreactors, these stages are balanced, and products of intermediate stages do not accumulate. However, under stress conditions (low pH, excess of ammonium, high organic loading rate, etc.), the system can become unbalanced, which leads to the accumulation of volatile fatty acids (VFAs). High VFA concentrations inhibit

microorganisms and can seriously impair or completely stop the process of methanogenesis. Acetic, propionic and butyric acids are the main VFAs that accumulate when AD is destabilized. In addition, depending on the process conditions, other metabolites such as ethanol, glycerol, lactate, etc., can accumulate [2–5]. In the absence of inorganic electron acceptors, VFAs are degraded by syntrophic consortia of microorganisms (except acetate, which can also be utilized by methanogenic archaea), in which VFA oxidation is carried out by fatty-acid-oxidizing bacteria and hydrogen formed is consumed by hydrogenotrophic methanogens, resulting in the very low concentration ($10^{-4}$ atm) that is needed for VFA degradation [6]. At the same time, high VFA concentrations can inhibit syntrophic processes. Therefore, it is necessary to search for effective syntrophs that are resistant to high VFA concentrations.

To date, several bacteria have been isolated and described that are able to convert fatty acids in syntrophy with methanogens; bacteria produce mainly hydrogen and acetate, which are utilized by methanogens to produce methane. Many of these belong to the class *Deltaproteobacteria* and the low G+C Gram-positive bacteria, including members of genera, e.g., *Syntrophus, Syntrophobacter and Desulfovibrio* [7]. Two other groups of microbes that perform syntrophic metabolism fall into the low G+C Gram-negative bacteria. One group is composed of species within the genera *Desulfotomaculum, Pelotomaculum and Sporotomaculum. Syntrophomonadaceae* comprises another group of microbes that perform syntrophic metabolism in the low G+C Gram-negative bacteria and includes species in the genera *Syntrophomonas, Syntrophothermus* and *Thermosyntropha* [7]. *Pelotomaculum schinkii, Pelotomaculum propionicum, Syntrophomonas zenderii* and *Pelotomaculum isophtalicum* belong to obligate syntrophic bacteria [8–11]. However, most of the studied syntrophs can live both syntrophically and non-syntrophically, depending on the growth substrates [7]. Unlike obligate syntrophs, many facultative syntrophs are more resistant to changes in environmental conditions in bioreactors, because they can shift to an alternative source of carbon and energy and therefore might be of interest for biotechnology.

As a result of anaerobic lipid hydrolysis, glycerol, saturated and unsaturated fatty acids are formed. Glycerol is not an inhibitory compound, unlike C12-14 saturated fatty acids and C18 unsaturated fatty acids, which may inhibit the process of methanogenesis [12–14]. *Klebsiella pneumoniae, Citrobacter freundii, Enterobacter agglomerans* and *Clostridium butyricum* are some of the microorganisms that can use glycerol as a growth substrate [15]. In this case, decomposition can occur through the reduction pathway (through the formation of 3-hydroxypropionaldehyde) and/or through the oxidative pathway (oxidation of dihydroxyacetone to acetate, for example, in *C. butyricum*). *Gelria glutamica* is an obligately syntrophic, glutamate-degrading bacterium which can grow on lactate and glycerol in pure culture [16]. Information on the syntrophic degradation of glycerol is limited. According to Quatibi et al. [17,18], representatives of the genus *Desulfovibrio* are capable of using glycerol as a substrate for syntrophic growth in co-culture with a hydrogenotrophic methanogen. In this case, the spectrum of the formed metabolites differs for different species of *Desulfovibrio*, as well as depending on the presence of sulfate in the medium. In the presence of sulfate as an electron acceptor, the final products of a pure culture of *D. carbinolicus* are 3-hydroxypropionate and sulfide. During syntrophic decomposition of glycerol, *D. carbinolicus*, when co-cultivated with *Methanospirillum hungatei* without sulfate, produces 3-hydroxypropionate and methane. *D. fructosovorans* in co-culture with *M. hungatei* forms 3-hydroxypropionate, methane and a very small amount of 1,3-propanediol when glycerol is decomposed. In pure culture with sulfate, *D. fructosovorans* oxidizes glycerol to acetate [18]. When *Desulfovibrio alcoholovorans* and *M. hungatei* are co-cultivated in a medium with glycerol as the only substrate, acetate and methane are formed. In a pure culture with sulfate in the medium, *D. alcoholovorans* oxidizes glycerol to acetate [17]. *Tepidanaerobacter syntrophicus* degrades glycerol (as well as lactate) in co-culture with *Methanothermobacter thermoautotrophicus* [19]. Within the *Thermoanaerobacter* genus, a few species could grow with glycerol in pure culture [20–23]. However, co-cultivation of these *Thermoanaerobacter* strains with a methanogenic partner, *Methanothermbacter* sp., enhanced glycerol conversion [24].

The formation of lactate occurs at the stage of fermentation of mainly sugars in the process of lactic acid fermentation (*Lactobacillus casei*, *Streptococcus lactis*, *Leuconostoc mesenteroides*, *Oenococcus oeni*) and mixed-type fermentation (*Thermoanaerobacter mathranii*, *Clostridium thermocellum*). Lactate has been suggested to be one of the most important intermediate compounds in the methanogenic degradation of carbohydrates in the thermophilic anaerobic wastewater treatment process [25]. Representatives of the genus *Desulfovibrio* are one of the key groups capable of syntrophic decomposition of lactate [26–29]. For syntrophic oxidation of lactate by *Desulfovibrio* spp., it is necessary to maintain a low concentration of sulfates; otherwise, sulfate reduction will occur. Microorganisms capable of syntrophic growth on lactate belong to the families *Thermoanaerobacteraceae* (*Tepidanaerobacter syntrophicus*) and *Desulfovibrionaceae* (*Desulfovibrio desulfuricans* and *D. vulgaris*) [19,26]. *Thermodesulfovibrio aggregans* and *Th. thiophilus* grow syntrophically on lactate together with *Methanothermobacter thermoautotrophicus* $\Delta H^T$ [30].

Recently, a representative of a new species and a new genus, *Biomaibacter acetigenes* SK-G1$^T$, was isolated from oily sludge samples at the Shengi oilfield in PR China [31]. This strain was able to grow in pure culture with a number of substrates including lactate, but it could not use glycerol. In that study, the potential of *B. acetogenes* to grow syntrophically with methanogens was not investigated. In this work, a new strain of *Biomaibacter acetigenes* SP2 was isolated from thermophilically digested sewage sludge of a municipal wastewater treatment plant. Besides the *Biomaibacter acetigenes* SK-G1$^T$ type of strain, our isolate is also a phylogenetic relative of *Tepidanaerobacetr syntrophicus* JL$^T$, which grows syntrophically on lactate and glycerol [19]. Here, we show that the strain *Biomaibacter acetigenes* SP2 is capable of syntrophic association with methanogens when growing on lactate and glycerol.

## 2. Materials and Methods

### 2.1. Medium

Pfennig's modified medium [32] was used for microbial enrichment and isolation. The medium contained 10 mL/L of solution #1 and solution #2, 2 mL/L of Lippert microelements solution [33] and 2 mL/L of Wolin vitamin solution [34]. Solution #1 contained (g/L) $NH_4Cl$–33, $MgCl_2.2H_2O$ (or $MgCl_2.6H_2O$)–33 (50), $CaCl_2.6H_2O$–33, KCl–33. Solution #2 contained $KH_2PO_4$–33 g/L. $Na_2S$ (0.5 g/L) and cysteine (0.5 g/L) were used as reducing agents, and $NaHCO_3$ (2.5 g/L) was used to maintain the buffer capacity. A resazurin solution (1 mL/L) was used as an indicator of anaerobic conditions. Yeast extract (0.5 g/L) was used as an additional source of growth factors. The pH of the medium was about 7.0.

### 2.2. Enrichment and Isolation of Strains SP1 and SP2

The source of strain isolation was thermophilically digested sewage sludge from Kuryanovskaya municipal wastewater treatment plant (Moscow, Russia). The moderately thermophilic (55 °C) syntrophic community resistant to high concentrations (170 mM) of butyrate was obtained by adaptation to gradually increasing concentrations of butyrate (from 10 to 170 mM) [35]. The initial idea was to isolate a butyrate-oxidizing syntrophic bacterium; therefore, a hydrogenotrophic methanogen in co-culture was also needed, best of all from the same consortium. The isolation of microorganisms and the characterization of the SP2 strain were started in 2016. The 120 mL bottles were used with 20 mL of liquid and 100 mL gas phase. For isolation of hydrogen-utilizing methanogens, the method of serial dilutions with $H_2/CO_2$ headspace and medium supplemented with vancomycin was used. It was noted that, in addition to a rod-shaped methanogen with slightly curved cells, another rod-like microorganism with thicker cells was present in the culture. To separate the two microorganisms, this culture was inoculated in roll tubes with 4% agar medium in different penicillin vials. The gas phase in the vial to isolate the methanogen was $H_2/CO_2$. To isolate the bacterium, crotonate (20 mmol/L) was added to the vials, and the headspace was purged by $N_2/CO_2$. Crotonate was chosen because the consortium grew on butyrate, and syntrophic butyrate-oxidizing bacteria are often able to grow in pure culture with crotonate. The medium contained 0.5 g/L of yeast extract. After the

colonies appeared, they were transferred to liquid medium with $H_2/CO_2$ or crotonate. As a result, on a medium with $H_2/CO_2$, a pure culture of a hydrogenotrophic methanogen was obtained, which was named strain SP1. On the medium with crotonate, a rod-shaped bacterium was isolated and named strain SP2. The purity of the cultures was confirmed by phase-contrast microscopy and sequencing of the 16S rRNA genes.

### 2.3. Morphology and Physiology of Strain SP2

The morphology and cell size of strains SP1 and SP2 during growth on various substrates was evaluated using phase contrast microscopes AxioLab.A1 and AxioImager.D1 (both Carl Zeiss, Jena, Germany). The transmission electron microscope JEOL 100C XII (Japan) was used to study the microstructure of cells, inclusions and flagella. The temperature range was tested from 15 to 80 °C using sucrose as a substrate. The pH range for growth was determined from 5 to 9 with the use of 10% HCl and NaOH. NaCl concentrations were tested in a range of 0–4%. Tolerance to oxygen of strain SP2 was tested by cultivation in standard medium with 20 and 50% of air in the gas phase. Substrates tested for growth of SP2 included: cellobiose, fructose, mannose, galactose, sucrose, glucose, rhamnose, cysteine, sorbate, cellulose, xylose, xylan, acetate, propionate, butyrate, methanol, ethanol, raffinose, ribose, lactose, arabinose, lactate, melibiose, glucosamine, pyruvate, formate, glycerol, crotonate, betaine (20 mmol/L each), malate (5 mmol/L), starch, casamino acids, yeast extract and peptone (2 g/L). Substrate tests were performed at the optimum temperature for growth of strain SP2 (55 °C). Several acceptors of electrons were tested: sulphate (10 mM), thiosulphate (10 mM), sulfite (2 mM), dithionite (10 mM) and colloidal sulfur with sucrose (20 mM) as electron donor.

### 2.4. Capability of Strain SP2 to Grow Syntrophically with a Methanogen

SP2 strain was tested for its ability to grow syntrophically by co-cultivation with strain SP1, a hydrogenotrophic methanogen, isolated from the same enrichment and closely related to *Methanothermobacter thermautotrophicus* $\Delta H$ (99.7% 16S rRNA gene identity). Methanogenic archaeon *Methanothermobacter thermophilus* strain M[T] (VKM B-1786) [36] and the syntrophic bacterium *Tepidanaerobacter syntrophicus* strain JL[T] (DSM 15584) [19], capable of syntrophically converting substrates such as lactate, glycerol and ethanol, were also used in syntrophic experiments as controls. *Methanothermobacter thermophilus* strain M[T] was obtained from VKM (Pushchino, Russia), and *Tepidanaerobacter syntrophicus* was purchased in DSMZ (Braunschweig, Germany).

Sodium acetate, sodium propionate, sodium butyrate, ethanol, methanol, glycerol, sodium lactate (20 mM each) and sodium benzoate (5 mM) were used as substrates. As a negative control, medium with appropriate substrates and strain SP2 without methanogen was used. To perform syntrophic experiments, pure culture of strain SP1 was cultivated on $H_2/CO_2$ until high cell density, and the gas phase of the bottles exchanged to $N_2$ before injection of substrates and inoculation of strain SP2 (2 mL).

### 2.5. Analytical Methods

The content of gaseous products ($H_2$, $CH_4$, $CO_2$) was determined using a Crystal 5000.2 gas chromatograph (Chromatec, Yoshkar-Ola, Russia) equipped with a thermal conductivity detector and a 1 m column filled with Carboxen-1000, 60–80 mesh (Supelco, Bellefonte, PA, USA). Separation was carried out isothermally at 140 °C. Gas carrier was argon. Ethanol, acetate and the other VFAs were determined using a Crystal 5000.2 gas chromatograph (Chromatec, Russia) equipped with a flame ionization detector and a capillary column ZB-WAXplus 30 m $\times$ 0.25 mm $\times$ 0.35 μm (Phenomenex, Torrance, CA, USA). The column was operated with temperature programming from 100 to 180 °C with a ramp rate of 10 °C/min. The gas carrier was nitrogen. Glycerol and lactate concentrations were analyzed using a Stayer HPLC (Aquilon, Moscow, Russia) equipped with a refractometric (Smartline 2300, Knauer, Berlin, Germany) and UV detector mounted in series. Separation was performed isocratically on an Aminex HPX-87H column (Bio-Rad, Hercules, USA).

Before injection, the samples of microbial suspension were freed from the cells by centrifugation at 10,000 rpm for 5 min. Clear supernatant was acidified by 5M $H_2SO_4$ to pH 2 and injected into the column.

Cellular fatty acids were determined by GC-MS of methyl ester derivatives prepared from freeze-dried biomass treated by anhydrous HCl/MeOH [37].

### 2.6. DNA Isolation, Sequencing and Phylogenetic Analysis

DNA was isolated by the modified method of alkaline DNA extraction by Birnboim and Doly [38] and the Wizard technology (Promega, United States). Polymerase chain reaction (PCR) and subsequent sequencing of the fragments of the 16S rRNA gene were performed with a universal primer system [39]. Amplification products were sequenced by the Sanger method [40] using a Big Dye Terminator v. 3.1 reagent kit (Applied Biosystems, Inc., United States) in an ABI PRIZM 3730 genetic analyzer (Applied Biosystems, Inc., United States) according to the manufacturer's instructions. Initial comparison of the newly obtained sequences with the sequences from the GenBank database was performed using the NCBI Blast software (http://www.ncbi.nlm.nih.gov/blast, accessed on 1 February 2023). The sequences were edited using BioEdit (http://jwbrown.mbio.ncsu.edu/BioEdit/bioedit.html, accessed on 1 February 2023). The fragments of the 16S rRNA gene sequences of more than 1400 bp were obtained for all of the isolated strains. The dendrograms of phylogenetic affinity were plotted using MEGA 4.0 [41].

### 2.7. Genome Analysis

Genomic DNA was extracted according to the method of Wilson [42], with minor modifications. Libraries were constructed with the NEBNext DNA library prep reagent set for Illumina, according to the protocol for the kit. The sequencing was undertaken using the Illumina HiSeq 1500 platform with 220 bp reads. Raw reads were quality checked with FastQC v.11.7 (https://www.bioinformatics.babraham.ac.uk/projects/fastqc/, accessed on 1 February 2023), and low-quality reads were trimmed using Trimmomatic v. 0.36 [43]. Subsequently, the quality-filtered reads were de novo assembled with SPAdes version 3.11.0 using the default settings [44].

## 3. Results

### 3.1. Isolation and Morphology of Strain SP2

A consortium adapted to a high concentration of butyrate (170 mM) [35] was used to isolate pure cultures of a methanogenic archaeon and a bacterium. Colonies of two morphotypes were obtained in roll tubes in 15 mL penicillin bottles on 4% agar medium supplemented with 0.5 g/L of yeast extract and $H_2/CO_2$ (80/20%) gas mixture in the headspace. The first morphotype was dark gray with a smooth lens shape and the second was fluffy greyish colonies. Morphologically similar greyish colonies were also formed in roll tubes in a Pfennig medium with sodium crotonate as a substrate with yeast extract, 4% agar and $N_2/CO_2$ in the headspace. Smooth lens-shaped colonies were transferred to liquid medium flushed with $H_2/CO_2$. Rod-shaped cells developed in liquid medium, and methane was formed as a product. This organism was identified as closely related to *Methanothermobacter thermautotrophicus* according to the 16S rRNA gene sequence (99.7% identity with the *Methanothermobacter thermautotrophicus* $\Delta H$ type of strain) and named strain SP1. Greyish colonies were transferred to fresh liquid medium with sodium crotonate as the substrate and yeast extract as a standard component of the medium. As a result, a pure culture of strain SP2 was isolated. This bacterium was closely related to *Biomaibacter acetigenes* strain $SK-G1^{T.}$ (99.9% sequence similarity).

As mentioned before, strain SP2 was isolated from a syntrophic butyrate-oxidizing consortium. Moreover, phylogenetic close relatives of the strain in the GenBank are the syntrophic bacteria *Tepidanaerobacter syntrophicus* strain JL [19] and *Tepidanaerobacter acetatoxidans* strain Re1 [45]. We supposed that this bacterium also might have the capability of syntrophic growth; therefore, we studied its physiology.

Cells of strain SP2 were rods 2 to 10 μm in length and 0.5 μm in diameter. Cells were single, in pairs and sometimes in short chains (Figure 1A). Cells were motile due to one or more lateral flagella (Figure 1B). Cells stained positively according to Gram and possessed a Gram-positive cell wall. Sometimes, they formed terminal spores.

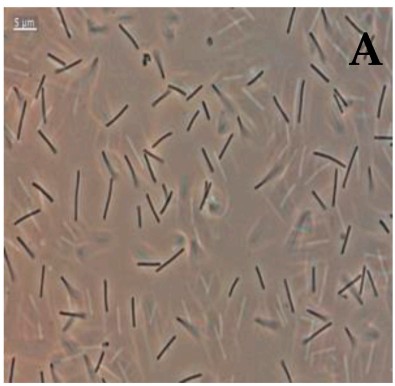 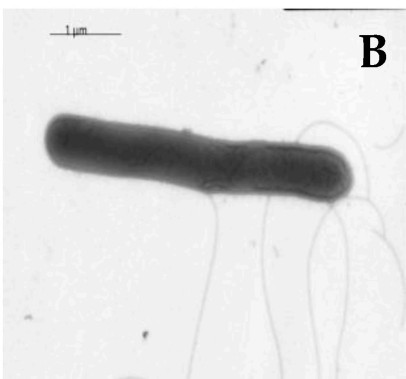

**Figure 1.** (**A**) Cell morphology of the SP2 strain grown on sucrose. Phase contrast micrograph. Bar: 5 μm. (**B**) Transmission electron micrograph of SP2 strain cell showing flagella. Bar: 1 μm.

Strain SP2 was capable of forming aggregates during syntrophic cultivation with hydrogenotrophic methanogens (Figure 2).

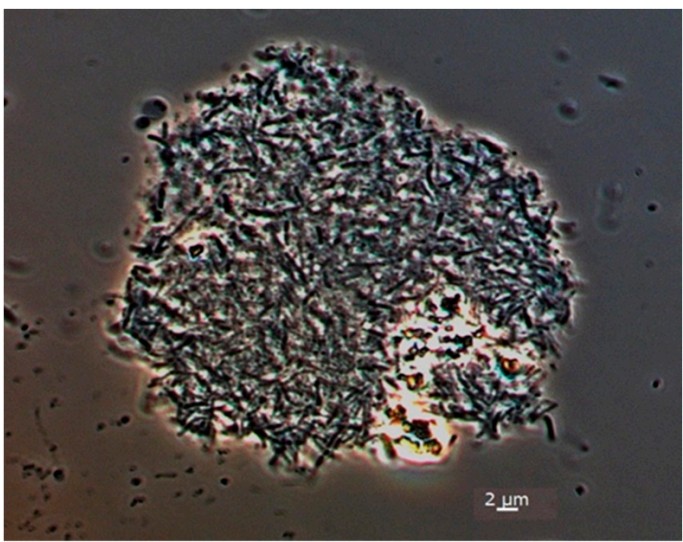

**Figure 2.** Morphology of cell aggregate *M. thermautotrophicus* and strain SP2 (phase contrast micrograph).

### 3.2. Physiology of Strain SP2

The isolated bacterium was an obligate anaerobe, a moderate thermophile with a growth range of 20 to 70 °C with an optimum at 55–60 °C and a neutrophile with growth at a pH of 3.5 to 8 with an optimum at 7.5. Growth was not observed at temperatures below 20 and above 70 °C and pH below 3.5 and above 8.0 for 30 days. Growth was observed in the range of NaCl concentrations up to 2.5%. The maximum growth rate was observed in the absence of extra NaCl. In pure culture, strain SP2 used the following substrates: yeast extract, cellobiose, fructose, mannose, galactose, sucrose and glucose. Yeast extract was not only a substrate but also a source of growth factors. Weak growth was observed on xylose, rhamnose, cysteine and casamino acids. No growth was observed on sorbate, cellulose, xylan, butyrate, acetate, propionate, malate, methanol, ethanol, peptone, raffinose, ribose, lactose, arabinose, lactate, melibiose, starch, glucosamine, pyruvate, formate, glycerol, crotonate and betaine. The products of glucose fermentation were acetate, ethanol, $H_2$ and

$CO_2$, which indicates that strain SP2 is a chemo-organoheterotroph with a mixed type of fermentation. The isolated bacterium used thiosulfate as an electron acceptor (to form $H_2S$) when grown on sucrose, and sulfur globules were formed inside the cells (Figure 3A,B). When grown with colloidal sulfur, sulfur globules were formed inside the cells without the formation of $H_2S$. Strain SP2 did not use sulfite, sulfate, dithionite and colloidal sulfur as an electron acceptor. After long-term cultivation of strain SP2 on sugars, it lost the ability to accumulate sulfur globules but not $H_2S$ formation.

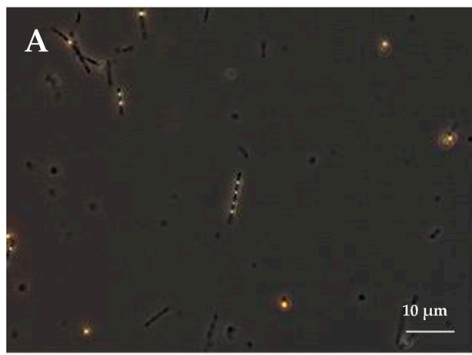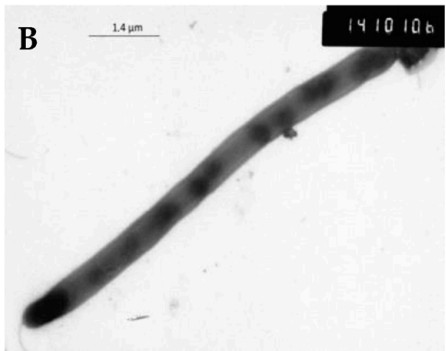

**Figure 3.** (**A**) Sulfur inclusions in SP2 strain cells grown on sucrose with thiosulfate as an electron acceptor. Phase contrast micrograph. Bar: 10 μm. (**B**) Micrograph of a cell of strain SP2 with sulfur inclusions localized inside the cell. Transmission electron micrograph. Bar: 1.4 μm.

*3.3. Syntrophic Growth*

Strain SP2 was capable of syntrophic growth on lactate and glycerol when co-cultured with the hydrogen-using methanogens *M. thermoautotrophicus* SP1 (the strain isolated in this research) or *M. thermophilus* $M^T$. Strain SP2 did not grow syntrophically on acetate, propionate, butyrate, methanol, ethanol and benzoate.

3.3.1. Syntrophic Degradation of Glycerol

When cultured with the hydrogenotrophic methanogen *M. thermautotrophicus* strain SP1, strain SP2 was capable of syntrophic conversion of glycerol. This methanogen was used as a putative syntrophic partner, since it was isolated simultaneously with strain SP2 from the same thermophilically digested municipal sewage sludge. The degradation of glycerol began without a lag phase, and the complete decomposition of glycerol occurred within 48 days (Figure 4A). Strain SP2 was also able to degrade glycerol when cultivated with *M. thermophilus* (Figure 4B).

As a control, *Tepidanaerobacter syntrophicus* $JL^T$ (capable of syntrophic growth on glycerol), which is the closest phylogenetic relative of the SP2 strain, was co-cultivated with *M. thermautotrophicus* SP1. Syntrophic degradation of 4 mM glycerol took place within 2 days (Figure 4C) in comparison with 3.5 mM/L within 5 days when strain SP2 was cultivated with *M. thermautotrophicus* SP1 and *M. thermophilus* M (Figure 4A,B).

3.3.2. Syntrophic Degradation of Lactate

When cultured with the hydrogenotrophic methanogen *M. thermautotrophicus*, strain SP2 was also capable of syntrophic degradation of lactate, preceded by a long lag phase. Complete degradation of 30 mM lactate occurred within 30 days (Figure 5A).

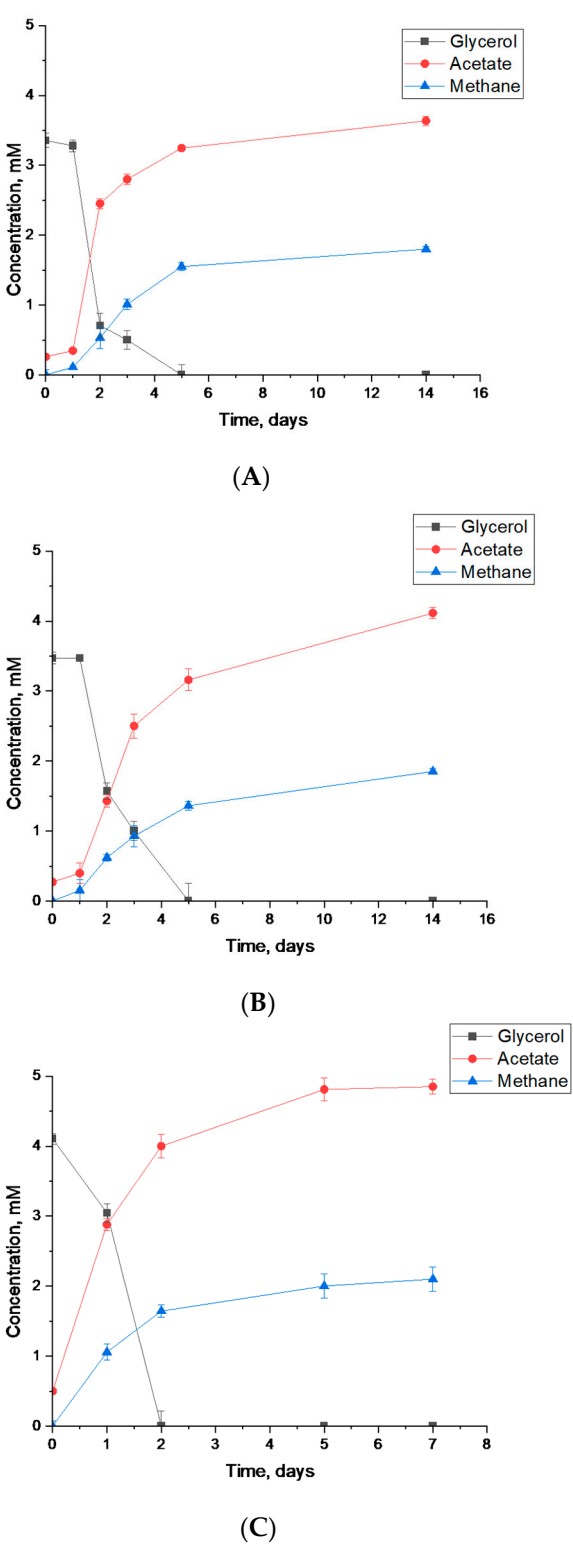

**Figure 4.** Syntrophic degradation of glycerol by strain SP2 and hydrogenotrophic methanogens *M. thermautotrophicus SP1* (**A**) and *M. thermophilus M$^T$* (**B**), and by *T. syntrophicus strain JL$^T$* and *M. thermautotrophicus* SP1 (**C**).

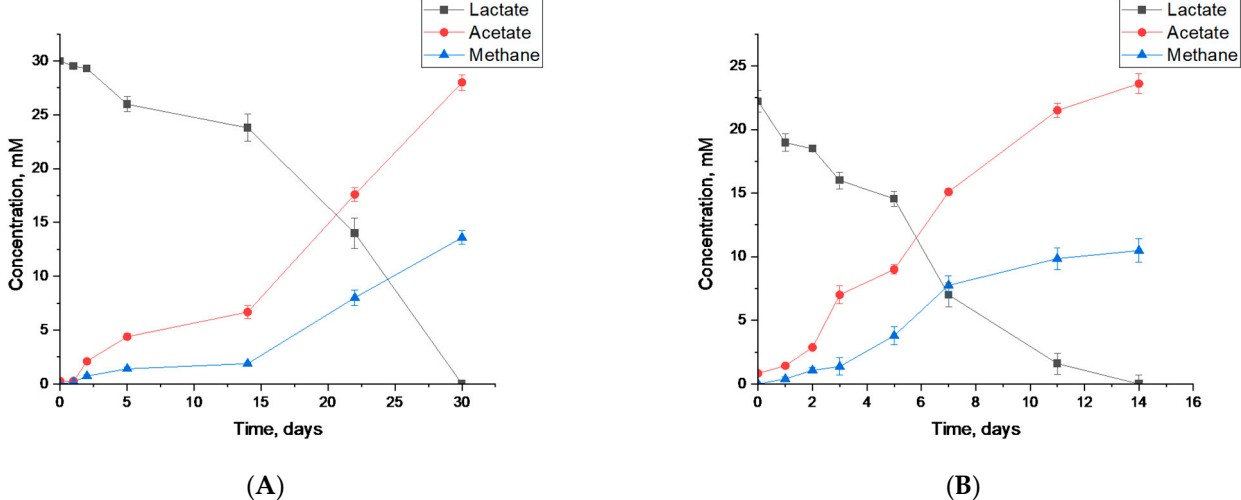

**Figure 5.** Syntrophic degradation of lactate by strain SP2 with *M. thermautotrophicus* SP1 (**A**) and by *T. syntrophicus* MT with *M. thermautotrophicus* SP1 (**B**).

*T. syntrophicus*, which is the closest phylogenetic relative of strain SP2, cultivated together with *M. thermautotrophicus*, was used as a control (Figure 5B). Syntrophic decomposition of 22 mM lactate occurred within 14 days; the rate was higher than during syntrophic decomposition of lactate by strain SP2 with the hydrogenotrophic methanogen *M. thermautotrophicus*. Thus, *T. syntrophicus* can degrade both glycerol and lactate at a higher rate than strain SP2.

Strain SP2 was not able to grow syntrophically on acetate (20 mM), propionate (20 mM), butyrate (20 mM), methanol (20 mM), benzoate (5 mM) or ethanol (20 mM) when cultivated with *M. thermautotrophicus* SP1 or *M. thermophiles* M[T].

### 3.4. Taxonomic Analysis

#### 3.4.1. Sequence of 16S rRNA Gene

A sequence of the 16S rRNA gene of the SP1 strain showed that it is phylogenetically close to *Methanothermobacter thermoautotrophicus* [36] (99.72% similarity). A sequence of the 16S rRNA gene of the SP2 strain demonstrated that its closest valid phylogenetic relatives are *Biomaibacter acetigenes* (99.9% identity) [31], *Tepidanaerobacter syntrophicus* (93.3%) [19] and *Tepidanaerobacter acetatoxydans* (93.0%) [45]. According to phylogenetic data based on the analysis of the 16S rRNA gene, strain SP2 together with *Biomaibacter acetigenes* SK-G1[T] formed a species-level clade near the root of the genus *Biomaibacter*. Strain SP2 is phylogenetically related to *Biomaibacter acetigenes* strain SK-G1[T] as revealed by comparison of the 16S rRNA gene (99.9% identity) and genome (ANI 99%, dDDH 90%) of both strains (Figure 6).

#### 3.4.2. Genome Features

A total of 2,566,901 reads were assembled into 101 total contigs. This represents approximately 160× average coverage of a total sequence length of 3,526,989 bp with a medium coating of 160×. The largest contig was 187,339 bp, with an N50 value of 91,597 bp and a G+C content of 43.1%. Annotations of contigs were carried out using the NCBI Prokaryotic Genome Annotation Pipeline (PGAP) [46], which identified 3573 genes, 3426 coding sequences, 88 pseudogenes and 49 tRNA genes. The GenBank accession number of the genome of strain LBB-42T is QSNL00000000. Phylogenetic analysis based on the 16S rRNA gene sequence and concatenated amino acid sequences of the core proteins was carried out using the maximum-likelihood methods, performed by IQ-TREE [47,48], with 1,000 bootstrap replicates [49].

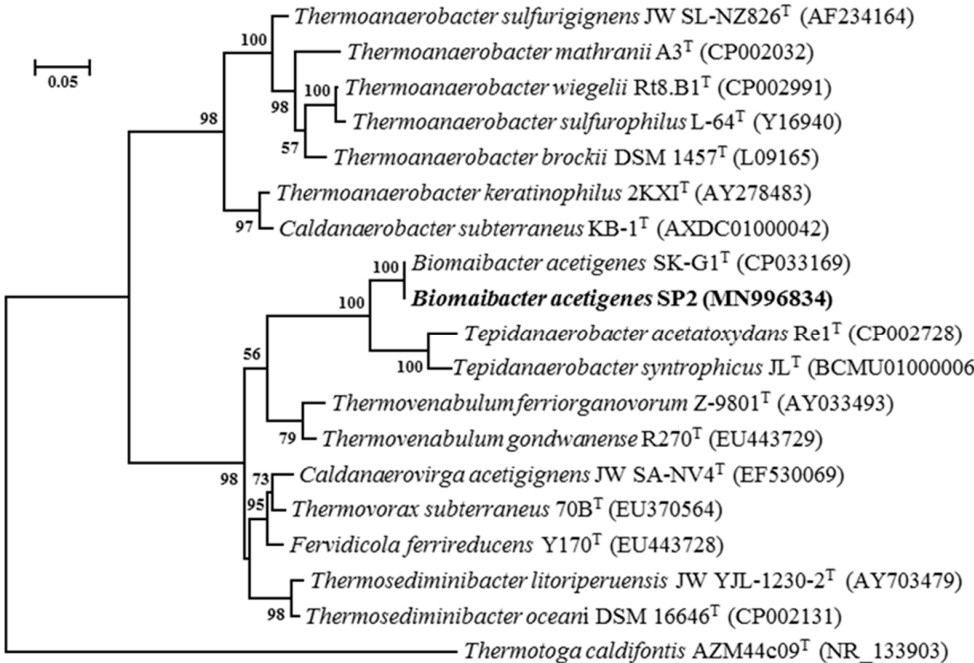

**Figure 6.** A maximum-likelihood phylogenetic tree based on 16S rRNA gene sequences (1407 nucleotide sites) reconstructed with the evolutionary model GTR + F + I + G4, showing the position of strain SP2 and closely related members of the order *Thermosediminibacterales*. Bootstrap values (>50%) are listed as percentages at the branching points. Scale bar, 0.02 substitutions per nucleotide position. The tree was rooted using *Thermotoga caldifontis* AZM44c09 as the outgroup. GenBank accession numbers for 16S rRNA genes are indicated in brackets.

### 3.4.3. Chemotaxonomic Characteristics

The main fatty acids of the cell wall of strain SP2 are saturated fatty acids: I-C15:0 (23.09%), ai-C15:0 (13.96%), C15:0 (12.4%) and C14:0 (9.26%). Unsaturated derivatives of C15:1 (1.62%) and DMA C15:1 (1.62%) are present in a substantially smaller amount (Table 1).

**Table 1.** Cellular fatty acids profile of strain SP2 in comparison with *Biomaibacter acetigenes* SK-G1[T] and *Tepidanaerobacter syntrophicus* JL[T].

| Name | ECL (HP-5MS) * | SP2 | *Biomaibacter acetigenes* SK-G1 | *Tepidanaerobacter syntrophicus* DSM 15584 |
|---|---|---|---|---|
| i-C11:0 | - | ND | ND | 1.3 |
| i-C12:0 3-OH | - | ND | ND | 1.2 |
| C12:0 3-OH | - | ND | 1.9 | ND |
| ai-C13:0 | 12.69 | 9.0 | ND | ND |
| C13:0 | 13.00 | 6.0 | 11.3 | ND |
| i-C14:0 | 13.62 | 5.1 | 3.1 | ND |
| C14:0 | 14.00 | 9.3 | 33.8 | 6.9 |
| Summed feature 1 (C13:0 3-OH or i-C15:1 H) | - | ND | 6.7 | ND |
| C15:1 ** | 14.39 | 1.62 | ND | ND |
| ai-C15:1 A | - | ND | 2.1 | 1.0 |
| i-C15:1 F | - | ND | ND | 16.6 |
| C15:1ω8c | - | ND | 6.2 | 3.5 |
| C15:1ω6c | - | ND | 1.4 | ND |



**Table 1.** *Cont.*

| Name | ECL (HP-5MS) * | SP2 | *Biomaibacter acetigenes* SK-G1 | *Tepidanaerobacter syntrophicus* DSM 15584 |
|---|---|---|---|---|
| i-C15:0 | 14.62 | 23.1 | 2.4 | 23.4 |
| ai-C15:0 | 14.70 | 14.0 | 4.9 | ND |
| C15:0 | 15.00 | 12.1 | ND | ND |
| DMA C15:1 | 15.11 | 1.5 | ND | ND |
| DMA C15:0 | 15.49 | 2.1 | ND | ND |
| i-C16:1 H | - | ND | 7.1 | ND |
| i-C16:0 | 15.62 | 1.0 | ND | ND |
| C16:0 2-OH | - | ND | ND | 2.3 |
| C16:1ω7 | 15.80 | 0.8 | ND | ND |
| C16:1ω9 | - | ND | ND | 2.9 |
| Summed feature 3 (C16:1ω7c or C16:1ω6c) | - | ND | 4.1 | 3.7 |
| C16:0 | 16.00 | 6.6 | 4.7 | 7.6 |
| C16:0 n-alcohol | - | ND | ND | 1.6 |
| DMA C16:0 | 16.49 | 1.2 | ND | ND |
| C17:0 10-methyl | - | ND | ND | 2.8 |
| C17:1ω6 | - | ND | ND | 3.1 |
| C17:1ω8 | - | ND | ND | 2.9 |
| i-C17:0 | 16.63 | 1.2 | ND | ND |
| ai-C17:0 | 16.72 | 1.1 | ND | ND |
| C17:0 | 17.00 | 1.3 | ND | ND |
| C18:1ω11 | 17.74 | 0.7 | ND | ND |
| C18:1ω9c | - | ND | ND | 6.3 |
| Summed feature 8 (C18:1ω7c or C18:1ω6c) | - | ND | ND | 2.4 |
| C18:0 | 18.00 | 2.4 | ND | 4.4 |

Note: * ECL, equivalent chain length for the HP-5MS column (low polar (5%-phenyl)-methylpolysiloxane phase). ** C15 branched unsaturated fatty acid. ND—not detected.

In Table 2, the physiology of strain SP2 is compared with close relatives.

**Table 2.** Characteristics of strain SP2 and members of related species and genera *Biomaibacter acetigenes* SK-G1[T] [33], *Tepidanaerobacter syntrophicus* JL[T] [19] and *Tepidanaerobacter acetatoxydans* Re1[T] [49]. –, negative; ±, weakly positive; +, positive; ND, no data available.

| Characteristics | Strain SP2 | *Biomaibacter acetigenes* SK-G1[T] | *Tepidanaerobacter syntrophicus* JL[T] | *Tepidanaerobacter acetatoxydans* Re1[T] |
|---|---|---|---|---|
| Source of isolation | Enrichment culture obtained from thermophilically digested sludge of the Kuryanoskaya wastewater treatment plant (Moscow, Russia) and adapted to high concentrations of butyrate. | Oily sludge at the Shengli oilfield in PR China. | Sludge of a full-scale thermophilic (55 °C) digester that decomposed municipal solid wastes (Japan). | A continuously stirred laboratory-scale reactor at 37 °C was fed with alfalfa silage and had high ammonium concentration (Sweden). |
| Cell morphology | Straight rods of different length | Slightly curved rods | Irregular rods | Irregular rods |
| Cells size, μm | 0.5–0.6 × 2.0–10.0 | 0.3–0.5 × 1.5–3.0 | 0.6–0.8 × 1.5–10.0 | 0.3–0.6 × 2.0–10.0 |
| Motility | + | + | − | + |
| Spore formation | + | + | − | + |
| Temperature range, °C | 20–65 | 35–65 | 25–60 | 25–55 |

**Table 2.** *Cont.*

| Characteristics | Strain SP2 | *Biomaibacter acetigenes* SK-G1[T] | *Tepidanaerobacter syntrophicus* JL[T] | *Tepidanaerobacter acetatoxydans* Re1[T] |
|---|---|---|---|---|
| Optimal temperature, °C | 55–60 | 55 | 45–50 | 37–47 |
| pH range | 3.5–8.0 | 6.0–8.5 | 5.5–8.5 | 4.5–9.5 |
| Optimum pH | 7.5 | 6.5–8 | 7.0 | 7.5 |
| Salinity, % (optimum, %) | 0–2.5 (0) | 0 | ND | ND |
| Fatty acid composition | iso-C15:0, anteiso-C15:0, C15:0, C14:0 | iso-C15:0, anteiso-C15:0, iso-C17:0, C16:1w9c and C18:1 w 9c, | iso–$C_{15:0}$, $C_{16:1}$ cis, $C_{15:1}$ | C18:1 w7c, C18:0, C18:1 w9c, anteiso-C17:0, C16:1 w7c |
| G+C content (mol%) | 43.1 | 43.9 | 38 | 38 |
| Yeast extract | + | + | − | ND |
| Casamino acids | + | ND | − | + |
| Tryptone/peptone | − | + | − | + |
| Pyruvate | − | + | − | + |
| Acetate | − | ND | − | − |
| Glycerol | − | − | − | + |
| Crotonate | − | ND | + | ND |
| Lactate | − | + | − | − |
| Glucose | + | + | + | + |
| Mannose | + | + | + | + |
| Galactose | + | + | + | + |
| Fructose | + | + | + | + |
| Arabinose | − | ND | ± | − |
| Sucrose | + | ND | + | − |
| Rhamnose | + | ND | ND | ND |
| Cellobiose | + | ND | ND | + |
| Raffinose | − | + | + | − |
| Xylose | + | + | + | − |
| Cysteine | + | ND | ND | + |
| Utilization of thiosulphate as electron acceptor | + (globules of sulfur are formed inside the cells) | + Fe(III), Mn(IV) in the presence of lactate | + | + in the presence of lactate |
| The main products of sugar fermentation | Acetate, ethanol, $H_2$, $CO_2$ | Acetate, $H_2$, $CO_2$ | Acetate, $H_2$, $CO_2$ | Acetate |
| Necessity of yeast extract as a growth factor | + | + | + | − |
| Capability of syntrophic growth | + (glycerol, lactate) | ND | + (glycerol, lactate, ethanol) | + (acetate) |

## 4. Discussion

According to phylogenetic analysis of the 16S rRNA gene, strain SP2 belongs to the species *Biomaibacter acetigenes*. Nevertheless, it has several phenotypic differences compared to the *Biomaibacter acetigenes* SK-G1[T] type of strain and members of the family *Tepidanaerobacteraceae*. The morphology of strain SP2 is characterized as straight rods of different length, while the morphology of *Biomaibacter acetigenes* SK-G1[T] is characterized as slightly curved rods. The *Biomaibacter acetigenes* type of strain can grow on lactate in pure culture, but strain SP2 cannot. Strain SP2 has a wider pH and temperature range than *Biomaibacter acetigenes* SK-G1[T]. Strain SP2 is more psychrotolerant than SK-G1[T]. It can grow at 20 °C in comparison to SK-G1[T], which does not grow at lower than 35 °C. Strain SP2 does not use peptone, pyruvate and lactate in pure culture, which are used by SK-G1[T]. Strain SP2 does not grow on crotonate as well as SK-G1[T].

Strain SP2 forms sulfur inclusions inside cells when growing in a medium with thiosulfate. It is known that some bacteria of the genus *Thermoanaerobacter* and *Caldanaerobacter* (for example, *T. sulfurigignens*, *C. subterraneus* subsp. *yonseiensis*), when grown with thiosulfate, are capable of accumulating sulfur globules both inside and outside cells [50–54]. The accumulation of sulfur globules inside the cells of *Thermoanaerobacter sulfurigingens* begins in the exponential growth phase, and the active formation and release of globules into the medium occurs in the stationary phase [54]. The molecular mechanism of the formation of sulfur globules inside cells is unknown.

The capability of strain SP2 to grow in syntrophy with hydrogenotrophic methanogens is a remarkable feature. Unfortunately, strain SK-G1[T] was not tested for syntrophic growth

ability [31]. Cultivation of strain SP2 with hydrogenotrophic methanogen results in the formation of aggregates. This phenomenon is very common among syntrophic microorganisms and serves to reduce the distance between syntrophic partners, thus enhancing the interspecies transfer of hydrogen [55]. Syntrophic degradation of lactate by strain SP2 paired with *M. thermautotrophicus* is consistent with the literature's data. The resulting products (acetate and methane) and their ratio correspond to the syntrophic decomposition of lactate by *Desulfovibrio vulgaris* with the methanogenic partners [56], following Equation (1):

$$2C_3H_5O_3{}^- + H_2O \rightarrow 2C_2H_3O_2{}^- + CH_4 + HCO_3{}^- + H^+ \tag{1}$$

The fatty acids content of strain SP2 is closer to that of the *Tepidanaerobacter syntrophicus* strain $M^T$, which is also able to grow syntrophically on glycerol and lactate, than that of SK-G1$^T$ (Table 2). It can be noticed that strains SP2 and *Tepidanaerobacter syntrophicus* $M^T$ originated from similar environments: SP-2 originated from the enrichment culture obtained from thermophilically digested sewage sludge of the Kuryanoskaya municipal wastewater treatment plant (Moscow, Russia) and strain M from the sludge of a full-scale thermophilic digester that processes municipal solid wastes (Niigata, Japan). Strain SK-G1$^T$ was isolated from sludge from an oil field.

## 5. Conclusions

Thus, as a result of this study, a new strain of *Biomaibacter acetigenes* SP2 was isolated from a methanogenic enrichment obtained from thermophilically digested sludge of the Kuryanoskaya wastewater treatment plant (Moscow, Russia). It was shown that, despite the phylogenetic similarity, strain SP2 has some physiological and morphological differences from the SK-G1$^T$ type of strain. Strain SP2 is capable of growing on glycerol and lactate syntrophically with the hydrogenotrophic methanogen *Methanothermobacter thermoautotrophicus*. The quantitative content of cell wall fatty acids is more similar to the phylogenetically distant *Tepidanaerobacter syntrophicus* DSM 15584 than to the phylogenetically identical strain *Biomaibacter acetigenes* SK-G1$^T$.

Facultative syntrophic microorganisms such as the newly isolated strain SP2 are able to switch to an alternative source of carbon and energy; consequently, they are more resistant to changes in environmental conditions. Therefore, their isolation and in-depth study are important for ecotechnologies for wastewater treatment. Future research may be directed towards using the novel strain for enhanced anaerobic treatment of wastewater rich in lactate and/or glycerol.

**Author Contributions:** Conceptualization, S.P.; methodology, S.P, V.K., N.K., A.N., D.K., T.K. and R.B.; validation, D.G.; formal analysis, A.N.; investigation, S.P, E.Z. and A.A.N.; resources, S.P. and Y.L.; writing—original draft preparation, S.P.; writing—review and editing, S.P. and Y.L.; visualization, N.K., E.Z. and D.G.; project administration, A.N.N.; funding acquisition, Y.L. All authors have read and agreed to the published version of the manuscript.

**Funding:** The study was supported by the Ministry of Science and Higher Education of the Russian Federation, grant No. 075-15-2022-1225. The work of T.K. and R.B. was supported by the Ministry of Science and Higher Education of the Russian Federation.

**Institutional Review Board Statement:** Not applicable.

**Informed Consent Statement:** Not applicable.

**Data Availability Statement:** Not applicable.

**Conflicts of Interest:** The authors declare no conflict of interest.

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
