# Peer review of "Syntrophic Growth of Biomaibacter acetigenes Strain SP2 on Lactate and Glycerol"

_fermentation, doi:10.3390/fermentation9060557_

Round 1
Reviewer 1 Report
This manuscript describes the discovery of a new strain Biomaibacter acetigenes Strain SP2 isolated from a wastewater treatment plant in Moscow, this new strain has been phenotypically and genotypically characterised, highlighting its ability to grow on Lactate and Glycerol, not reported for a similar strain.
The research has been carried out and designed properly. Results are interesting and the manuscript well presented. There are a several minor mistakes that need to be addressed before its publication. See below.
Specific comments below
L-22 –“ 55°C optimum)”, in results authors say 55-60 C, be more consistent
L59 it says “bacteria have been isolated and described that are able to convert fatty-acids in syntrophy with methanogens” can authors expand a bit and mention what the products this bioconversion are.
Results - Be more concise, unnecessary information in this section, that could be better placed in results/discussion section
L130-131 – “Solution â„–1 contained (per L):” – remove per L and just add g/L in each of the components.
Section 2.1 – add the pH of the final media
L141 – “The initial idea was isolation of a butyrate-oxidizing” – it better to say “was to isolate a butyrate-oxidizing…”
L144 – ml – typo
L-146-148 – this sentence should not be here
L-178-181 – this sentence should not be here
L-233-243 – this sentence should not be here better in results
L-246 - reference, keep the same format
L-286 – “Yeast extract was not only a substrate, but also a source of growth factors.” This sentence is too generic and unnecessary. This is well described for so many organisms.
L-309 – “was capable of syntrophic conversion of glycerol.” – better to say was assimilate or metabolise glycerol.
Figure 4 – units in Y axis are wrong, either mmol/L or mM. Same for Figure 5
Line 320 - 3.5 mM/L – wrong units
L395 – Thermoanaerobacter – in italic
Conclusion – avoid bullet points.
Good
Author Response
This manuscript describes the discovery of a new strain Biomaibacter acetigenes Strain SP2 isolated from a wastewater treatment plant in Moscow, this new strain has been phenotypically and genotypically characterised, highlighting its ability to grow on Lactate and Glycerol, not reported for a similar strain.
The research has been carried out and designed properly. Results are interesting and the manuscript well presented. There are a several minor mistakes that need to be addressed before its publication. See below.
Response: We thank the Reviewer for the positive perception of our work, constructive suggestions and comments.
Specific comments below
L-22 –“ 55°C optimum)”, in results authors say 55-60 C, be more consistent
Response: Corrected
L59 it says “bacteria have been isolated and described that are able to convert fatty-acids in syntrophy with methanogens” can authors expand a bit and mention what the products this bioconversion are.
Response: Done
Results - Be more concise, unnecessary information in this section, that could be better placed in results/discussion section
Response: Thanks for your comment; some text (particularly from section 2.7) has been moved to the "Results" section.
L130-131 – “Solution â„–1 contained (per L):” – remove per L and just add g/L in each of the components.
Response: Corrected
Section 2.1 – add the pH of the final media
Response: added
L141 – “The initial idea was isolation of a butyrate-oxidizing” – it better to say “was to isolate a butyrate-oxidizing…”
Response: Corrected
L144 – ml – typo
Response: Corrected
L-146-148 – this sentence should not be here
Response: the following sentence “It was noted that, in addition to a rod-shaped methanogen with slightly curved cells, another rod-like microorganism with thicker cells was present in the culture.” could be moved to the results, but in this case we have to move the other part of the paragraph as well, because the integrity of the rest of the text will be lost.
L-178-181 – this sentence should not be here
Response: we believe the following sentence “SP2 strain was tested for its ability to grow syntrophically by co-cultivation with strain SP1, an hydrogenotrophic methanogen, isolated from the same enrichment and closely related to Methanothermobacter thermautotrophicus ∆H (99.7% 16S rRNA gene identity).” should be left in the Materials and Methods section as it is part of the description of the enrichment and isolation procedure for SP1 and SP2 strains.
L-233-243 – this sentence should not be here better in results
Response: agree, moved to results
L-246 - reference, keep the same format
Response: Corrected
L-286 – “Yeast extract was not only a substrate, but also a source of growth factors.” This sentence is too generic and unnecessary. This is well described for so many organisms.
Response: Here we meant that the yeast extract was a permanent component of the mineral medium. In addition, we tested it as a growth substrate, but in a higher concentration.
L-309 – “was capable of syntrophic conversion of glycerol.” – better to say was assimilate or metabolise glycerol.
Response: here we cannot agree with you, because the SP2 bacterium cannot metabolize glycerol in pure culture, only in syntrophic cooperation with hydrogenotrophic methanogen.
Figure 4 – units in Y axis are wrong, either mmol/L or mM. Same for Figure 5
Response: Corrected
Line 320 - 3.5 mM/L – wrong units
Response: Corrected
L395 – Thermoanaerobacter – in italic
Response: done
Conclusion – avoid bullet points.
Response: Corrected
Reviewer 2 Report
The authors should be commended for such good work. The method covered what was required for the work. which was also detailed and appropriate. the results were well presented and explained. the sample can be said for the discussion and conclusion.
Minor comments are:
Line 143: were store at -80 during this period? since it was cultured in 2016. did you verify the microorganism before use?
Line 260: did you also send SP2 for sequencing? is this the one that was isolated in 2016?
Author Response
The authors should be commended for such good work. The method covered what was required for the work. which was also detailed and appropriate. the results were well presented and explained. the sample can be said for the discussion and conclusion.
Response: Many thanks to the Reviewer for the positive assessment of our work.
Minor comments are:
Line 143: were store at -80 during this period? since it was cultured in 2016. did you verify the microorganism before use?
Response: We changed this sentence to “The isolation of microorganisms and the characterization of the SP2 strain were started in 2016.”
Line 260: did you also send SP2 for sequencing? is this the one that was isolated in 2016?
Response: we did not store this strain after isolation at -80°C. We only meant that this work was started and almost completed much earlier than the paper on the description of the type strain Biomaibacter acetigenes SK-G1T (Zhang et al., 2019) was published and therefore we had to change the concept of our paper.